# Beyond Haemostasis and Thrombosis: Platelets in Depression and Its Co-Morbidities

**DOI:** 10.3390/ijms21228817

**Published:** 2020-11-21

**Authors:** Benedetta Izzi, Alfonsina Tirozzi, Chiara Cerletti, Maria Benedetta Donati, Giovanni de Gaetano, Marc F. Hoylaerts, Licia Iacoviello, Alessandro Gialluisi

**Affiliations:** 1Department of Epidemiology and Prevention, IRCCS NEUROMED, 86077 Pozzilli, Italy; izzi.epigenomica@gmail.com (B.I.); nina.tirozzi@gmail.com (A.T.); chiara.cerletti@moli-sani.org (C.C.); mbdonati@moli-sani.org (M.B.D.); giovanni.degaetano@moli-sani.org (G.d.G.); licia.iacoviello@moli-sani.org (L.I.); alessandro.gialluisi@gmail.com (A.G.); 2Center for Molecular and Vascular Biology, Department of Cardiovascular Sciences, University of Leuven, 3000 Leuven, Belgium; 3Department of Medicine and Surgery, University of Insubria, 21100 Varese, Italy

**Keywords:** depression, platelets, cardiovascular disease, psychiatric comorbidities, neurodegenerative disorders, platelet distribution width, gender

## Abstract

Alongside their function in primary haemostasis and thrombo-inflammation, platelets are increasingly considered a bridge between mental, immunological and coagulation-related disorders. This review focuses on the link between platelets and the pathophysiology of major depressive disorder (MDD) and its most frequent comorbidities. Platelet- and neuron-shared proteins involved in MDD are functionally described. Platelet-related studies performed in the context of MDD, cardiovascular disease, and major neurodegenerative, neuropsychiatric and neurodevelopmental disorders are transversally presented from an epidemiological, genetic and functional point of view. To provide a complete scenario, we report the analysis of original data on the epidemiological link between platelets and depression symptoms suggesting moderating and interactive effects of sex on this association. Epidemiological and genetic studies discussed suggest that blood platelets might also be relevant biomarkers of MDD prediction and occurrence in the context of MDD comorbidities. Finally, this review has the ambition to formulate some directives and perspectives for future research on this topic.

## 1. Introduction

Platelets are increasingly considered a putative bridge linking mental, immunological, and coagulation-related disorders, alongside their function in primary haemostasis and thrombo-inflammation. Patients with complex neurological and neuropsychiatric disorders such as schizophrenia, Parkinson and Alzheimer’s diseases have already been described with altered platelet function several decades ago [1,2,3,4,5,6]. Since then, numerous studies have given better insight into those early observations, by deciphering the cellular, molecular and functional similarities between platelets and neurons [7,8,9,10,11], and they have set the basis for the use of blood platelets as a working model to study several neurological diseases [3,8,9,12,13,14,15,16].

Platelets not only mirror neurons in several aspects, but directly contribute to the pathophysiology of those diseases affecting them in several ways. We will focus on the link between platelets and the pathophysiology of major depressive disorder (MDD), one of the leading causes of the disability and disease burden worldwide. MDD presents as a persistent low mood, with associated changes in behavior, cognition, sleep and appetite, impaired social and occupational functioning, increased risk of self-harm or suicide [17,18]. MDD is also associated with increased mortality due to several comorbidities, especially cardiovascular disease (CVD). Moreover, depression shows a high comorbidity with many other disabling conditions, including neurodegenerative, neuropsychiatric and neurodevelopmental disorders. With this review we first aim at providing functional evidence for a platelet pathophysiological involvement in depression by highlighting the intrinsic platelet characteristics in drug-free depressed patients. Additionally, we present here an overview of the current state of the art in epidemiological and genetic studies linking MDD and its main co-morbidities with platelets, in addition to providing some original data on the epidemiological link between platelet parameters and depression symptoms.

## 2. Platelet “Bridge” Proteins Linking Thrombosis and Inflammation with Depression

Despite their different embryonic origin [15,19], platelets and neurons share several characteristics. Platelets mimic the stable synaptic structure between neurons, i.e., where they interact with each other [4]. With neurons, they share the complex molecular machinery that regulates granule trafficking [7], controlling the calcium-dependent release reaction of stored agonists, after activation stimulation in platelets and the neurotransmitter release following an action potential in neurons [20]; they both have similar secretory vesicles in terms of content, storing molecules such as serotonin or 5-hydroxytryptamine (5-HT), dopamine, epinephrine, glutamate, gamma-aminobutyric acid (GABA), calcium, adenosine 5′-diphosphate (ADP) and Adenosine 5′-triphosphate (ATP). Platelets and neurons also share a number of proteins that include serotonin transporters and receptors [8,19,20] as well as some markers originally known as neuron-specific such as reelin, amyloid precursor protein (APP) and brain-derived neurotrophic factor (BDNF) [10,21,22].

Several studies have identified an association of the (epi)genetic variability of genes encoding some of the key players in depression etiology with both mood disorders and thrombo-inflammatory conditions such as CVD. Those studies have been inspired by the functional role that neuron- and platelet- shared proteins have in platelets themselves. In many instances, these genes encode shared proteins between neurons and platelets including the serotonergic pathway, cannabinoid receptor 1 and *BDNF* [23,24,25,26,27,28]. *BDNF* is probably the most studied of them as one of its specific variants (Val66Met, G to A, rs6265) has been consistently reported to influence the predisposition to CVD associated with depression [25,27,29]. Although (epi)genetic studies identifying novel genes with pleiotropic effects on both platelets and neurons in the context of depression are missing, there is increasing functional evidence that links some of these proteins to both depression and thrombo-inflammation. In this section of the review, we will go over this evidence by describing the role of four of the most relevant platelet proteins involved in platelet function and depression.

### 2.1. Serotonin (5-HT)

Discovered in 1935 by the Italian pharmacologist Vittorio Erspamer [30], 5-HT is a monoamine neurotransmitter implicated in numerous neuropsychological processes [31], including the pathophysiology of MDD [32,33]. Owing to the role of serotonin in platelets, the idea was born to use platelets as an experimental system for neurons [34,35,36], stemming from the observation of its specific uptake by platelets and from its release mechanisms and pharmacological regulation in animal models [37,38,39,40,41]. Several studies have subsequently confirmed that platelets are an adequate model of serotonin storage and release from serotonergic neurons, under both physiological and pathological conditions [32,34,35,36,42,43]. Platelets are the main serotonin reservoir in the bloodstream [44] resembling its vesicular storage in neurons [45]. Taken up from the circulating blood via the serotonin transporter (SERT or SLC6A4) [46,47,48,49,50], serotonin is stored in dense granules via the vesicular monoamine transporter 2 (VMAT2 or SLC18A2) [6,51,52] from where it can be released after platelet stimulation [53,54] to take part in platelet activation and acute inflammatory responses. Platelets express the serotonin receptors 2A (5-HT2A) and 3A (5-HT3A): platelet binding of serotonin activates PLCβ leading to intracellular Ca^2+^ release that stabilizes platelet activation. Autocrine serotonin signaling therefore contributes to platelet integrin αIIb/β3 activation and to an increased surface expression of P-selectin [55]. Important similarities have been observed between 5-HT2A receptor characteristics [56,57,58] or SERT structure [59] in the brain cortex and in platelets. These similarities have led to a handful of studies in platelets of depressed patients with or without other co-morbidities, mainly focusing on serotonin metabolism. Depressed patients showed up-regulation of the 5-HT receptor, decreased serotonin transporter binding, and decreased rate of 5-HT uptake [60]. Increased density of the 5-HT2A receptor has been detected in platelets of depressed suicide victims [61,62,63]. In depressed patients, SERT showed a decreased number of serotonin binding sites [33,64,65], and a decreased platelet SERT maximal velocity [66]. SERT serotonin transport was found to be enhanced in patients with depression [67] and was identified as the main pharmacological target of selective serotonin reuptake inhibitors (SSRIs), which prevent serotonin uptake in platelets [59].

More recently, the modulation of innate and adaptive immunity by platelet serotonin has been implicated in the regulation of inflammation [68,69,70,71,72]. The latter is also supported by the evidence that 5-HT receptors are expressed on numerous inflammatory cells besides platelets [73]. Platelet 5-HT is able to regulate both P- and E-selectin expression on murine endothelial cells, thereby influencing important inflammatory processes such as the rolling and adhesion of neutrophils [74,75,76,77,78].

### 2.2. Brain Derived Neurotrophic Factor (BDNF)

BDNF is a secretory protein regulating the development and function of neural circuits [79,80,81], expressed in both central and peripheral nervous systems [82]. Despite also being expressed in a number of peripheral tissues and cell types, platelets are the main source and storage cell of peripheral BDNF, representing 90% of total blood BDNF [22,83,84,85]. In line with this evidence, the *platelet count* (Plt) appears to be the factor that is most associated with serum BDNF concentrations [86,87]. BDNF is expressed and stored in human megakaryocyte α-granules together with platelet factor 4 (PF4) [88] and is released by platelets at the site of injury during platelet aggregation [85,89,90], impacting thrombosis [88,91]. This process occurs through activation of the protease-activated receptor 1 (PAR1) during thrombin stimulation [92]. Interestingly, BDNF stored in platelets not only derives from megakaryocytes, but also from cells in the brain and other organs that can, therefore, modulate BDNF levels in the circulation [89].

Altered BDNF levels have been found in patients with depression [93,94] and its co-morbidities [95,96,97]. Platelet granules containing BDNF were decreased in MDD [98,99]. Moreover, while about half of the BDNF platelet content is released upon platelet activation [83], some evidence suggests that in depressed patients the release of BDNF seems to be independent of platelet reactivity [84].

### 2.3. Reelin

Reelin is a neuronal protein regulating brain development, synaptic plasticity and memory formation [100,101,102]. It is not only expressed in the central nervous system, but also in a number of peripheral organs and in blood [103]. On the one hand, reelin is able to foster vascular inflammation via leukocyte–endothelial adhesion, thereby promoting atherosclerosis [104]. On the other hand, it upregulates thrombin generation and the formation of fibrin clots leading to blood coagulation [105]. Reelin can also be actively released following platelet activation thereby inducing Akt, Erk, Syk, and 1-phosphatidylinositol-4,5-bisphosphate phosphodiesterase gamma-2 (PLCγ2) activation through its contact with glycoprotein Ib (GPIb), the APP and the ApoER2 receptor [106]. It also supports platelet binding to collagen and glycoprotein VI (GPVI)-dependent RAC1 activation, PLCγ2 phosphorylation, platelet activation and aggregation [107], and it is a positive regulator of platelet spreading on fibrinogen [108].

### 2.4. Amyloid β Peptides (Aβ Peptides)

Aβ peptide overproduction is a common feature of both Alzheimer’s disease and chronic central nervous system diseases including depression [109,110,111,112]. At the same time, Aβ peptides are strongly proinflammatory, proapoptotic, and proatherogenic molecules with Ab1-40 being abundantly expressed in endothelial cells [113], vascular smooth muscle cells [114], macrophages [115], monocytes and platelets [113,116,117].

Aβ peptides are proteolytic fragments of APP, an integral membrane protein [118,119]. The *APP* gene produces 3 major splice variants, with the APP770 being specifically expressed in megakaryocytes and platelets [120,121]. In platelets, APP is stored in α-granules from where it is released upon platelet activation. On the one hand, released APP is able to participate in haemostasis by influencing coagulation, thrombosis and bleeding [122,123,124,125,126]. On the other hand, amyloid Aβ peptides can trigger platelet activation, adhesion and aggregation [21] through a number of different pathways: they can interact with CD36 and GP1bα and activate p38 MAPK/COX1 pathways, leading to thromboxane A2 (TxA2) release [127]; they can bind to integrin αIIbβ3 and trigger the release of ADP [128]; they are able to bind the PAR1 thrombin receptor and stimulate Ras/Raf, PI3K, P38MAPK, and cPLA2 and TxA2 formation and release [129]; they can act as ligand for the platelet activation receptor GPVI [130]; they can induce protein kinase C (PKC) activation and tyrosine phosphorylation through a NOX-dependent pathway [131]; they may cause platelet shape change and granule release through activation of the small GTPase RhoA and phosphorylation of myosin light chain kinase leading to cytoskeletal reorganization [21]. There is some evidence that the amyloidogenic pathway is also regulated at the post-transcriptional level by a number of platelet miRNAs including the ones regulating fibrinogen [132], the β site APP cleaving enzyme 1 (*BACE1*) and APP expression [133].

Few studies performed on both human and mouse platelets, have shown that anti-platelet treatments, such as clopidogrel and aspirin, are able to interfere with APP/Aβ peptide generation and function in platelets and/or neurons [128,134,135,136].

## 3. MDD and Platelets: Further Evidence of a Link

### 3.1. Epidemiological Studies

The relation between platelets and depression has been deeply investigated, both through epidemiological and—less often—through statistical genetics approaches. Some studies reported increased platelet activation in individuals with depression, compared to healthy controls. Musselman et al. observed an enhanced baseline platelet activation and responsiveness in patients affected by MDD, as suggested by the increased expression of platelet αIIbβ3 and P-selectin [137], while Pinto and colleagues [138] demonstrated an impairment of L-arginine-nitric oxide signaling in platelets of depressed compared to healthy subjects. Morel-Kopp et al. [139] reported a direct association of MDD with a higher number of CD62- and CD63-positive platelets and excitability, which were attenuated by a 6-month treatment with anti-depressants, in line with previous findings on platelet secretion in response to collagen binding [140]. An increased oxidative stress and hyperaggregability were observed in platelets of MMD cases compared to controls [141], as well as a higher content of serotonin, interleukin 1β, PF4 and CD40 ligand (CD40L) [142]. In line with this evidence, a recent longitudinal study on young males reported mental stress to be associated with increased and prolonged proinflammatory platelet bioactivity: while exposure to chronic stress led to an increased number of CD63^+^ platelets, acute stress was associated with alterations of CD62P^+^, CD63^+^, PAC-1^+^ platelets and of platelet–leukocyte aggregates [143].

Other epidemiological studies investigated the relation between MDD and platelets by making use of platelet parameters commonly tested, like mean platelet volume (MPV) and Plt. A positive association between MPV and MDD was reported in a Turkish population sample (N = 2286, 287 cases) [144], and later replicated in a study comparing 103 MDD patients and 106 controls [145], as well as in a hospital-based study (90 cases vs. 49 controls) [146]. However, these studies revealed contrasting evidence of association between Plt and MDD status: while Bondade and colleagues observed an increased Plt in depressed patients [146], Cai et al. found no statistical evidence supporting that finding [145]. They reported a positive association between MDD and plateletcrit (PCT), i.e., the product of MPV and Plt, which represents the total mass occupied by platelets in the blood [145]. Platelet parameters have also been studied with reference to MDD treatments: in a small study comparing 15 MDD patients under escitalopram therapy—one of the most used SSRI treatments—and 17 healthy controls, treated patients exhibited a significant reduction in both MPV and Plt, which were instead higher than in controls at baseline [147]. Another study comparing 31 patients with life-long recurrent depression treated with SSRIs and 31 matched healthy controls, reported significantly higher MPV, platelet distribution width (PDW, an index of size variability of circulating platelets) and platelet-to-larger cell ratio (P-LCR; i.e., the proportion of large platelets with volume >12 fL, which represents an index of platelet size useful in the diagnosis of thrombocytopenia) in depressed participants [148]. Although a direct link between PDW variability and platelet function has not yet been fully established [149], this evidence suggests once again that lower platelet activation and function may be a feature of depression, along with lower platelet and blood plasma serotonin, and lower platelet reactivity. Moreover, studies on collagen- and epinephrine-induced aggregation and the percentage of spiny and discoid platelets also suggested a lower platelet reactivity as a potential feature of depression [148]. In line with this evidence, in a comprehensive analysis of the relation between low-grade inflammation and mental health in a large Italian population cohort (the Moli-sani study; N = 12,732), our group identified a significant positive association between continuous depressive symptoms and PDW [150]. This association survived conservative adjustments for several sociodemographic, health and lifestyle covariates, suggesting the existence of shared genetic underpinnings between depressive status and platelet size variability [150].

### 3.2. Genomic Studies

The epidemiological findings reported above represent a robust rationale to investigate the shared basis between depression risk and platelet variability also at the genetic level. In a large genome-wide association study (GWAS) of blood cell measures (N_max_~170,000) [151], the authors performed a multivariable Mendelian randomization (MR) analysis on platelet parameters and MDD risk, so as to investigate potential causative effects of the former—assumed as exposure—on the latter, taken as an outcome (see [152,153,154,155] for overviews on MR rationale, assumptions and techniques). This revealed no significant causal effect of platelet parameters on MDD, although MPV and PDW showed marginally significant effects, which in any case did not survive correction for multiple testing [151]. More recently, Wray et al. reported a large GWAS on depression, with 135,458 cases and 344,901 controls, identifying 44 independent loci associated with MDD risk [156]. The authors performed genetic correlation analyses to investigate significant genetic overlaps—better defined as single nucleotide polymorphism (SNP)-based co-heritability [157,158]—between MDD risk and other different disorders and traits, including Plt and MPV. They observed no significant genetic correlations between depression and platelet parameters [156]. Following these studies, we recently revisited the link between platelet parameters and MDD risk using summary statistics from these two studies, combining genetic correlation and MR analysis [149]. Beyond using GWAS summary statistics of platelet parameters from a sample size much larger than the one used before (~166,000 vs. ~67,000 participants) [151,159], for the first time we investigated PDW in relation to MDD risk at the genetic level, reporting a significant positive genetic correlation between the two traits [149]. However, MR analyses revealed no evidence of a causal relationship between PDW and MDD, probably due to the low number of instrumental variants overlapping between the two studies; further investigations are warranted in this perspective, using summary statistics from even larger GWAS as they become available. In spite of the interesting co-heritability between PDW variability and MDD risk at the genomic level, the specific genes and variants at its basis need to be elucidated, and dedicated efforts are under way for this purpose.

## 4. Sex and Its Neglected Role in the Common Soil between Depression and Platelet Pathophysiology

Sex effect on the platelet-MDD association has been largely neglected in previous studies [144,145,146]. To this end and for the purpose of this review, we made used of the Moli-sani cohort to investigate the influence of sex on the link between depression and platelet indices. Specifically, we performed generalized linear models (glm) using depressive symptoms (standardized PHQ9-6 scale) as outcome and platelet parameters (standardized Plt, MPV and PDW) as exposures—stratified by gender, adjusted for age, lifestyles and chronic prevalent conditions, as in Gialluisi et al. [150]. Associations in women were further adjusted for variables potentially influencing hormonal status such as current menopause status, use of oral contraceptives and use of hormonal replacement therapy, so as to avoid potential confounding effects [160,161,162]. Moreover, we carried out a glm in the total sample (N = 12,732) [150], further adjusted for sex and including an interaction term with the latter factor, for each of the platelet parameters tested. This original analysis (reported in Table 1) revealed a substantial lack of association of Plt with depressive symptoms in both sexes, while MPV showed a significant negative association, but only in women. Interestingly, a positive association of PDW with depressive symptoms was present in both groups, with the effect size in women being two-fold higher than in men. However, in the total sample, we observed no significant interaction effect of PWD and sex on depressive symptoms. Of note, while potential confounders modifying hormonal status in women did not affect associations between platelet parameters and depressive symptoms within the Moli-sani study, sex hormones may still play a role in this association and further systematic analyses are warranted to rule out a potential influence of these variables, when hormone titers will be available within the cohort.

## 5. MDD Comorbidities and Platelets

Depression and depressive symptoms are frequently comorbid with cardiovascular disease, neurodegenerative, neuropsychiatric and neurodevelopmental disorders [163]. The latter have important clinical implications since depression may dramatically contribute to worsen those diseases and in general have an impact on overall health [164]. Indeed, comorbid depression has been associated with worse prognosis and increased mortality [165,166] and with a higher risk of developing other diseases later in life [167]. Several mechanisms have been proposed to explain the co-occurrence of depression with the comorbidities mentioned above, including treatment-induced morbidity, behavioral and psychological factors, but also underlying biological processes [163]. In this section, we will review the studies showing the implication of blood platelets as a possible common underlying marker of depression in comorbidity with cardiovascular disease and in major neurodegenerative, neuropsychiatric and neurodevelopmental disorders. A summary of shared associated platelet markers among MDD and its comorbidities is reported in Table 2.

### 5.1. Cardiovascular Disease

Depression and depressive symptoms have long since been established as important risk factors for cardiovascular disease related mortality. Clinical depression was identified as a significant risk factor for mortality in patients with coronary heart disease (CHD) [222] or myocardial infarction (MI) [223], and in middle-aged men with stroke [224]. Patients with depression had a two- to four-fold higher risk of death after a cardiac event [225,226], and showed higher risk of CVD mortality between 6 and 18 months following MI [227,228]. MDD patients are also at a higher risk of dying after a congestive heart failure (CHF) [229,230] and they have an increased CVD morbidity [225]. In patients with an acute MI, depression was found to be a risk factor for cardiac mortality independent of cardiac disease severity [231,232], and was associated with an increased risk of death in young women with coronary artery disease (CAD) [233]. In another study, depressed patients showed a 77% increased risk of all-cause mortality, 10 years after percutaneous coronary intervention [234]. A direct relationship between depressive symptoms and all-cause mortality was also found in the Framingham Heart Study [235] and in the Moli-sani [236] cohorts.

Clinical depression was not only associated with CVD and all-cause mortality, but was also shown in numerous reports to be a leading risk factor for CVD occurrence. Several studies published over the last 30 years have additionally indicated a bidirectional co-morbidity between MDD and CVD occurrence. In a meta-analysis of 124,509 individuals across 21 studies Nicholson and colleagues identified an 80% increased risk for developing coronary artery disease in association with depression [237]. Depressed patients had a two- to four-fold risk of developing CVD at some point in their lifetime [238,239,240,241,242,243], and different epidemiological studies have highlighted that patients suffering from ischemic heart disease show a high incidence of depression [244,245,246,247]. On the one hand, MDD was also associated with an increased risk of developing stroke (Hazard Ratio 1.45) [248]; on the other hand, 30% of stroke survivors developed MDD according to two meta-analyses [249,250]. Therefore, a bidirectional causality link between these disorders has been hypothesized. Comorbidity between depression and cardio-metabolic traits has been reported in a number of studies [251].

Beside potential behavioral explanations for this increased risk such as medication noncompliance, cigarette smoking and physical inactivity, other biological factors including a proinflammatory state, endothelial dysfunction and/or platelet activation have been considered [242,252,253]. Increasing evidence has pointed to a specific role for platelets in influencing the CVD-MDD comorbidity [254,255,256]. First of all, a higher platelet aggregability has been considered as a marker of patients with both CVD and MDD, as platelet hyperactivity could explain both pathological phenotypes [255,257]. As highlighted previously, higher platelet aggregability is a signature of depressed patients without cardiovascular events. Depressed patients display increased platelet serotonin receptor concentrations [189,190] and abnormally low platelet SERT levels [192] which would result in elevated serotonin concentration in the bloodstream. This would in turn lead to abnormal platelet aggregation in atherosclerotic arteries [258,259]. Indeed, elevated blood levels of serotonin are predictive of CAD and ischemic cardiac events in patients with suspected CAD [260], and in vitro experiments have demonstrated higher platelet aggregability in CVD/MDD patients [199,218,219,261]. In addition to that, post-myocardial depressed patients showed abnormal whole blood and platelet serotonin levels [199], and depressed CVD patients have a higher serotonin receptor density [191]. Anxiety, often accompanying depression (see below), has been shown to be a predictor of adrenaline and serotonin-dependent platelet reactivity in CAD patients [262]. All these findings were supported by the evidence that an SSRI-treatment decreased platelet aggregation and activity in CAD patients [263,264], and they led to several clinical trials to evaluate the health effects of SSRIs versus placebo or no antidepressants in patients with CHD and depression [265].

Because higher platelet aggregability was not always a consistent finding in patients with the MDD/CVD comorbidity [254,266,267], other studies have instead focused on measuring platelet activity by means of platelet activation markers or metabolization. Some studies have looked at PF4 and β-thromboglobulin (β-TG) levels in depressed patients with or without CAD [218,219,220,221] or alternatively in cerebrovascular disease [268] and non-depressed matched controls. In general, both PF4 and β-TG were higher in depressed compared to non-depressed CAD patients [218,219,220], as well as in depressed CAD patients compared to CAD- and depression-free controls [221]. In a recent study [269], the activity of platelet-dependent NAD and NADP dehydrogenases has been measured as read-out of platelet metabolism in patients with acute coronary syndrome (ACS) with or without concomitant anxiety-depressive disorder (ADD) followed up for one year for recurrent cardiovascular complications. Among the patients that developed cardiovascular complications over the follow-up period, the ones with the concomitant ACS and ADD had lower NADP–MDH activity compared to controls and to ACS patients without ADD [269].

Very few studies have tried to investigate whether platelet indices (among several markers) represent good risk factors or markers of MDD and CVD comorbidity. Increased PLR and MPV were shown to be predictive biomarkers of the development of post-stroke depression in acute ischemic stroke patients [181,182,270]. Except for one study where no significant association was found between Plt and somatic and cognitive depression symptoms concomitant with MI [174], other similar reports focusing on platelet indices and CVD comorbidities of depression are lacking.

### 5.2. Neurodegenerative Diseases

Common neurodegenerative diseases linked to the accumulation of neurotoxic protein aggregates are usually diagnosed when the disease is already at an advanced stage of neurodegeneration [12,271]. This makes it very important to identify potential biomarkers that are easy to measure and that could predict the incident risk of these disorders, e.g., as circulating biomarkers [272]. Platelets are, in different ways, associated with the pathophysiology of neurodegenerative disorders. First, they have a crucial role in the metabolism and storage of dopamine, Aβ peptides and APP [10], as previously discussed. This led the way to a handful of functional platelet studies briefly reviewed here below (most relevant human studies) and more extensively elsewhere [273]. More recently, the link between neurodegenerative disorders and classical blood platelet parameters like MPV, Plt and PDW has been investigated also at the epidemiological level, in a relatively limited number of studies. Indeed, platelet indices are easy-to-measure and standardized across laboratories and could be used as good prediction and/or prognosis markers of neurodegenerative risk. Based on these premises, we will focus on the most prevalent neurodegenerative disorders due to the accumulation of neurotoxic protein aggregates, i.e., Alzheimer’s disease (AD), Parkinson disease (PD) and amyotrophic lateral sclerosis (ALS), reviewing epidemiological and genetic studies as well as the main functional analyses performed in cohorts of patients.

#### 5.2.1. Alzheimer’s Disease (AD)

Both functional platelet markers and platelet indices have been reported in association with AD clinical presentation.

Platelet activation markers have been investigated in AD patients in several studies, often leading to inconclusive and inconsistent findings. Higher platelet CD62P (P-selectin) expression, platelet and platelet–leukocyte mixed aggregates were found in AD patients [217]; this evidence contrasts with that of another study showing increased soluble P-selectin levels but no different CD62P expression in resting platelets, as well as lower levels in platelets stimulated by thrombin receptor activating peptide 6 (TRAP-6) [274]. In four studies by Prodan and colleagues, increased numbers of pro-coagulant protein-coated platelets were detected in early-stage AD patients compared to controls [275,276,277,278].

Contrasting evidence has been reported regarding platelet serotonin in AD patients. Lower [194,195,196], normal [197,198], and increased [193] platelet serotonin uptake were all shown in AD patients. Measuring intraplatelet serotonin levels also led to inconsistent findings as well [200,201,202,203]. The role of neuron and platelet serotonin in depression and AD has been further explored in a review by Meltzeret et al. [279].

Based on more consistent and replicated data, the Aβ peptides together with monoamine oxidase B (MAO-B) have been considered as more relevant AD biomarkers in platelets [280]. The functional role of platelet Aβ peptides in AD has been largely investigated in animal models so that AD can be partly conceived as a thrombo-hemorrhagic disorder [281,282]. Aβ peptides are released at the site of vessel damage and contribute to the progression of the disease by regulating vascular amyloid deposits after platelet activation [21,123,124,283]. Several independent studies reported changes in platelet APP in AD patients [284,285,286,287,288,289,290,291].

Independent epidemiological studies have reported consistent associations between AD risk and platelet indices, in particular MPV and PDW. In a study comparing 150 Chinese vascular dementia (VaD), 110 AD patients and 150 non-demented controls, MPV and PDW were significantly lower in demented subjects and in AD, compared to vascular dementia patients [187]. Moreover, a positive correlation was observed between MPV/PDW levels and the mini-mental state examination (MMSE) score, a test commonly used to assess cognitive performance [187]. This finding was replicated in an independent Chinese study comparing AD, mild cognitive impairment (MCI) and cognitively healthy subjects (N = 120 for each group), which revealed a positive association between cognitive MMSE score, MPV and PDW. Authors also observed significantly lower MPV and PDW in AD patients vs. both MCI and control subjects, and in MCI vs. controls [188]. Another independent Chinese study of 92 AD patients and 84 age and sex-matched normal controls reported several alterations in quality and quantity of blood cells in demented patients, including a higher MPV and a lower PDW [183]. A higher MPV in AD patients was also found in a Turkish study comparing 89 AD, 93 PD patients and 104 healthy controls, although this did not correlate with MMSE performance [175]. In the same study, no significant associations were observed for Plt.

These findings suggest that platelets might have a role in AD pathology, which may also share genetic underpinnings. So far, two attempts were made to investigate this hypothesis. One used an MR analysis on platelet count (Plt), MPV, and PDW (used as exposure) vs. AD (modeled as outcome), which revealed no significant causal association [151]. However, this technique may suffer from low power since it is usually based on a low number of variants [292]. More recently, we attempted to estimate the co-heritability between these platelet parameters and AD risk based on common genetic variants, through linkage disequilibrium score regression, using information from more than one million variants in the genome and GWAS summary statistics from large genetic studies [151,293]. However, we found no evidence of significant genetic correlations for any of the platelet parameters tested. This suggests that the significant associations observed in previous epidemiological studies may be mainly due to shared environmental influence between platelet parameters and AD risk, and that common genetic influences on these traits are likely very limited, at least those of common SNPs [294].

#### 5.2.2. Parkinson Disease (PD)

Platelet studies have been performed in the context of PD etiopathology reporting changes in the ultrastructure, mitochondrial dysfunction, increase in glutamate level and abnormal morphology [295]. However, also in the case of PD, evidence is inconsistent across studies. For example, PD platelets have been shown to have a reduced mitochondrial complex I activity [211,212,213] an evidence not corroborated by other studies [214,215]. Platelet MAO-B activity, the enzyme responsible for complex I inhibition in mitochondria [296], was increased in PD patients vs. controls in some reports [205,206,207,208], but was unchanged in others [209,210].

A handful of more consistent epidemiological and genetic studies investigated potential links with common platelet indices. A study comparing 80 Turkish PD patients and 80 healthy controls revealed no significant difference in Plt nor in MPV between the two groups [176]. However, MPV showed a significant negative correlation with a PD severity scale—the Hoehn and Yahr score—suggesting this could be a biomarker of PD progression in the later stages of PD [176]. In partial concordance with this finding, increased MPV has been reported in a study comparing Turkish AD and PD patients vs. controls (see above), with MPV being negatively correlated with the Hoehn and Yahr score [175]. Of interest, PD patients showed significantly higher MPV compared to AD patients, as well, while Plt showed no significant difference across the compared groups [175]. Although some of the above-mentioned findings have been interpreted as suggestive of a link between inflammation (indicated by lower MPV) and stage of neurodegeneration (indicated by a PD progression score) [175], little is known on the pathophysiology of these disorders to make strong inferences, suggesting caution in the interpretation of these findings. Alternative hypotheses to explain the link between platelets and PD onset drew attention to mitochondrial dysfunctions in platelets, although contrasting findings have been reported on with this regard [214].

Genomic—rather than genetic—approaches to clarify this link have been attempted more recently, with the aim of investigating the presence of shared genetic underpinnings between platelet parameters and PD risk. Nalls et al. [297] tested genetic correlations between Plt/MPV and PD risk through linkage disequilibrium (LD) score regression [157,158], using summary statistics from the largest PD case–control GWAS meta-analysis carried out so far (involving ~56,300 PD cases and ~1.4 million controls), reporting no significant genetic correlations [297]. More recently, we extended this analysis to PDW and observed a significant positive genetic correlation with PD risk, suggesting the existence of a genomic overlap based on common genetic variants and indicating PDW as a new potential biomarker for PD [294]. In spite of the interesting finding, which is in line with the implication of PDW in neurodegenerative disorders [10,183,187,188] and in comorbid disorders like major depression [149], it should be noted that previous epidemiological studies reported negative associations between PDW and the risk of cognitive impairment [183,187,188], which is co-morbid and partly shares biological bases with PD [298]. At the genetic level, our group only identified a trend of (negative) correlation between PDW and AD risk [294]. Again, our limited knowledge of these disorders and of PDW itself does not currently allow us to further disentangle these aspects, although we are working towards identifying genes with pleiotropic influences on platelet variation and function and neurodegenerative risk.

#### 5.2.3. Amyotrophic Lateral Sclerosis (ALS)

Another main neurodegenerative disease showing an increased risk of depression—both before and after diagnosis—is amyotrophic lateral sclerosis [299]. A further connection between the above-mentioned disorders comes from evidence implicating glutamate transmission in MDD etiology and treatment [300] and altered glutamate transportation in ALS patients [301]. For example, ALS patients show a notable reduction in high-affinity glutamate uptake in platelets, compared with normal controls and chronic neurologic disorder patients [301]. Moreover, a selective loss of glial glutamate transporter GLT-1 (EAAT2) has been reported in brain and spinal cord from sporadic ALS patients [302]. This evidence suggests a systemic impairment of platelet functionality and specifically of glutamate transport in ALS, as does the finding that platelets from ALS patients show a 37% increase in expression of glutamine synthetase [303], although in this study normal expression of the glutamate transporter EAAT2 was observed. Of interest, platelet stimulation with thrombin resulted in an approximately sevenfold increase in glutamate uptake [303]. Although data were not statistically different between ALS patients and healthy controls, this trend suggests that glutamine synthetase could represent a good peripheral marker of ALS [303]. In addition to the above-mentioned studies, both qualitative and quantitative variations in ALS platelets and platelet mitochondria have been observed, including a heterogeneous distribution of granules, formation of vacuoles, blebs, pseudopodia, as well as perturbance of mitochondrial membrane potential, mitochondrial depolarization, apoptosis and lesser intra-mitochondrial granules [216]. A recent study reported higher levels of the 43 kDa TAR DNA-binding protein (TDP-43) in platelets from sporadic ALS patients that increase with disease progression [304]. This protein is implicated in the onset of the pathology by contributing to the formation of insoluble intracellular inclusions in neurons, and is present in the majority of ALS cases [305,306].

As for classical platelet indices, we are not aware of any study investigating potential associations with ALS risk.

### 5.3. Neuropsychiatric and Neurodevelopmental Comorbidities of MDD

Among psychiatric comorbidities of MDD, one of the most investigated conditions with reference to platelet parameters is represented by panic disorder (PanDis). Indeed, serotonergic agents have been reported to relieve panic symptoms [307]. Some of these studies reported evidence of increased MPV in PanDis patients vs. healthy controls [177,184,185], while others reported significant associations with an opposite direction of effect [168,169,178]. Those works which analyzed PDW reported higher values for cases [168,169,185], except for one detecting no significant differences [178]. Likewise, for Plt, contrasting results were reported: higher count in PanDis patients [168,169] vs. no significant difference between cases and controls in [177,178].

Other studies have focused on generalized anxiety disorder (GAD), a related disease characterized by a persistent and chronic anxiety state. In the first (retrospective) study specifically focusing on GAD patients, these showed higher MPV and lower Plt, but no significant differences in PDW and PCT, compared to healthy controls [173]. In partial concordance with these findings, GAD patients exhibited a higher MPV and Plt, as did MDD patients, in a study comparing depressive and anxiety disorder patients vs. healthy controls in a hospital setting [146].

Patients with acute stress have been reported to have greater platelet aggregation [308,309].

Post-traumatic stress disorder (PTSD) patients show increased platelet reactivity and aggregation in response to ADP and epinephrine stimulation [310]. A four-year longitudinal analysis of US veterans (N = 746) revealed no significant associations between Plt and the risk and course of PTSD, although Plt showed a significant association with a poorer course of depression [311].

Neuropsychiatric conditions, still, present differences in platelet parameters also when compared among themselves. Wysokiński and Szczepocka [312] compared platelet parameters like Plt, MPV and P-LCR among 2377 subjects including schizophrenia, depression, bipolar disorder and mania patients and identified several differences among these groups, for all the markers tested.

Among neurodevelopmental disorders, alterations in platelet parameters have been associated with important conditions like autism spectrum disorder (ASD) (as reviewed in [9]) and attention deficit hyperactivity disorder (ADHD) [171,172,186]. For ASD, anomalies have been mainly detected in platelet activation mechanisms and transport of granules [9], and a single study reported a mild increase in Plt for ASD cases and their siblings compared to controls [170]. For ADHD, although studies are still very limited, more consistent evidence of increased MPV and PCT in cases vs. control has been reported [171,172,179,180,186]. Of interest, MPV has been associated with inattention symptoms within ADHD cases, as well as with anxiety symptoms—along with Plt—in a specific subgroup of ADHD patients [172]. More recently, also significantly increased PDW in ADHD children compared to controls has been reported, beyond increased levels of Plt, MPV and PCT [171,179,180,186].

Once again, PDW presents as a novel and interesting candidate platelet biomarker for neuropsychiatric risk, although its functional and prognostic meaning remains to be elucidated.

## 6. Conclusions

To summarize, the independent lines of evidence reported here suggest that platelet pathophysiology has strong implications in the occurrence of MDD and of its related comorbidities (Figure 1) and they support the view that platelets reflect a circulating form of neurons [11]. However, several aspects in this fascinating hypothesis need yet to be disentangled.

First of all, the shared genetic bases between platelet variability and MDD, despite some first attempts [149,294], still needs to be elucidated through larger genetic epidemiology studies. The latter should take into account important features such as gender differences that are known to be linked to both platelet variability [313] and to the occurrence of MDD risk [314]. More studies that consider the occurrence of certain MDD comorbidities in specific population subgroups (i.e., women and pregnant women, children, aging population) are still lacking but are utterly important for further investigations.

**Figure 1 ijms-21-08817-f001:**
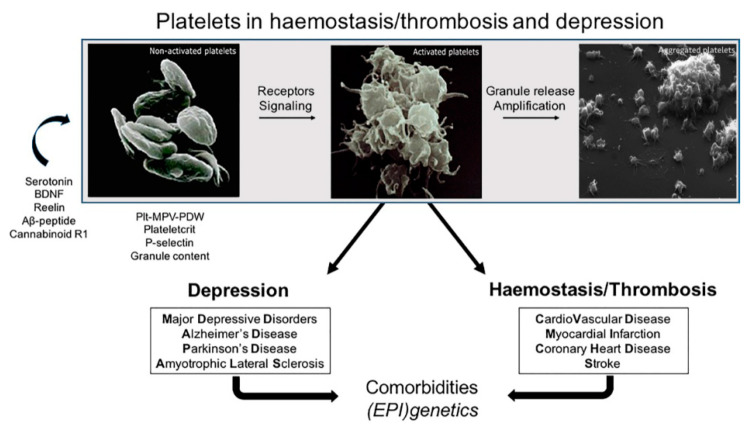
Shared platelet functionality between cardiovascular and neurological disease. Activation of platelets via specific receptors, signaling cascades and degranulation allows platelets to enhance tissue inflammation and to stimulate thrombotic complications, initiating and aggravating cardiovascular malfunction. A number of specific mediators of neurological disease are also capable of influencing platelet function and platelet activation appears to be linked to the development of those neurological and psychiatric diseases, in essence attributed to brain degeneration, i.e., markers of platelet activation may reflect the stage of mental disease progress. Several of these mediators are encoded by genes whose (epi)genetic variability has been involved in both cardiovascular and mental disease. Hence, various comorbidities of cardiovascular and mental disease may occur, because of the involvement of platelets in both disorders. Therefore, commonly measured indices of platelet mass, number and their activation, which are useful to characterize platelet functionality, seem useful biomarkers of the platelet concomitant involvement in both cardiovascular and neurological disorders. Plt: platelet count, MPV: mean platelet volume, PDW: platelet distribution width, MDD: major depressive disorder, AD: Alzheimer’s disease, PD: Parkinson disease, ALS: amyotrophic lateral sclerosis. Images of non-activated/activated and aggregated platelets were modified with permission from ref. [315] and [316], respectively.

Second, and related to the first point, the influence of sex on the relationship between depression risk/symptoms and platelet parameters needs to be clarified, following-up on the relevant evidence of moderate and interactive effects which we have reported here for the first time. Additional studies are warranted to verify whether the differential associations in women and men between platelet activation parameters and MDD results from intrinsic sex-specific (hormonal) differences influencing platelet activation or also from MDD-related factors.

Third, epidemiological and genetic studies increasingly identify PDW as an interesting yet functionally poorly understood candidate biomarker to test in terms of prediction of MDD and related neuropsychiatric/neurodegenerative risk in the future. The largely concordant evidence reported here that links PDW to MDD, ADHD, PanDis, as well as to cognitive traits and disorders like AD (Figure 2), is an incentive to further investigate this marker and its clinical implications. A particular topic of investigation may be its different associations and genetic correlations with PD and AD, two common neurodegenerative disorders with a partly shared biological basis.

Lastly, platelet serotonin might not only be important in platelet activation and subsequent involvement in platelet-related pathologies that are co-morbid with MDD, but also for its role as an epigenetic modulator in the serotonylation, a covalent posttranslational modification occurring at the level of histones thereby influencing gene expression [317]. Indeed, epigenetics has been hypothesized to play a major role in the etiology of depression [318,319,320]. In this view, serotonylation could represent a platelet-mediated link between serotonin and depression onset, a hypothesis which warrants further investigation.

As it appears from these open issues, much remains to be done to clarify the relationship between platelets, depression and its comorbidities in detail. This review tries to lay out at least some pieces of the puzzle for investigators in the field, having the ambition to expand into future research on this topic.

## Figures and Tables

**Figure 2 ijms-21-08817-f002:**
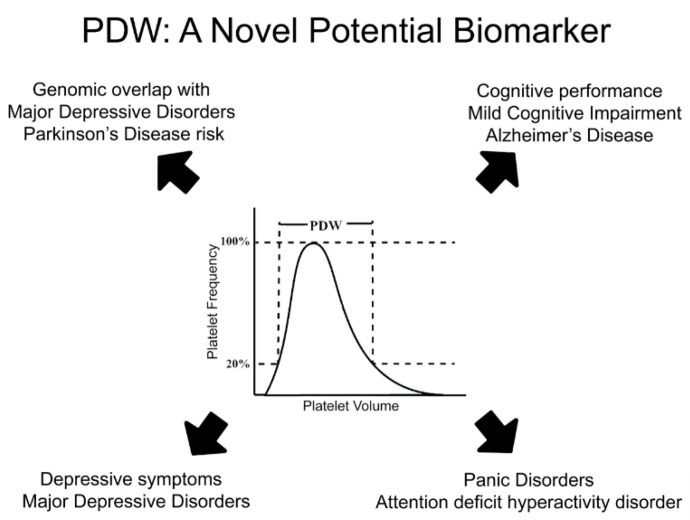
PDW: a novel potential biomarker for depression, its neurodegenerative and psychiatric comorbidities. Platelet distribution width (PDW) represents and index of platelet volume variability in a subject. It has been associated with both depressive symptoms and major depression, but also with neurodegenerative and psychiatric comorbidities like Alzheimer’s disease, mild cognitive impairment, attention deficit hyperactivity disorder and panic disorder. Beyond epidemiological evidence, genomic studies identified consistent co-heritability based on common genetic variants between PDW and depression, as well as between PDW and Parkinson disease risk. Overall, this evidence suggests PDW as a very promising candidate biomarker for MDD and its comorbidities to investigate in the future.

**Table 1 ijms-21-08817-t001:** Influence of sex on the association between platelet parameters and depression.

Platelet Parameter	Effect	Beta	SE	P	Setting (N)
Plt	Plt only	0.004	0.013	0.77	Women (6421)
	Plt + hormone-related variables ^a^	0.005	0.013	0.70	
	Plt only	−0.008	0.011	0.48	Men (6311)
	Interactive (Plt-by-sex)	0.018	0.017	0.30	ALL (12,732)
**MPV**	**MPV only**	**−0.034**	**0.014**	**0.01**	**Women (6421)**
	**MPV + hormone-related variables ^a^**	**−0.034**	**0.014**	**0.01**	
	MPV only	−0.015	0.011	0.16	Men (6311)
	Interactive (MPV-by-sex)	−0.025	0.017	0.14	ALL (12,732)
**PDW**	**PDW only**	**0.046**	**0.014**	**1 × 10^−3^**	**Women (6421)**
	**PDW + hormone-related variables ^a^**	**0.048**	**0.014**	**6 × 10^−4^**	
	**PDW only**	**0.021**	**0.010**	**0.04**	**Men (6311)**
	Interactive (PDW-by-sex)	0.029	0.017	0.09	ALL (12,732)

Sex-stratified associations between common platelet parameters (platelet count (Plt), mean platelet volume (MPV), platelet distribution width (PDW)) and depressive symptoms (PHQ9-6) and sex-by-platelet interaction analysis in the Moli-sani cohort. ^a^ Associations further adjusted for binary variables (yes/no) specifying current menopause status, use of oral contraceptives and of hormonal replacement therapy in women. Significant associations are reported in bold.

**Table 2 ijms-21-08817-t002:** Overview of shared platelet biomarkers across major depressive disorder (MDD) and its comorbidities.

Platelet Parameter	Depression Related Clinical Phenotype
**Plt**	↑ in MDD [146], PanDis [168,169], GAD [146], ASD [170], ADHD [171,172]
	↓ in GAD [173]
	No association with MDD [145], Depressive Symptoms with MI [174], AD [175], PD [175,176], PanDis [177,178], ADHD [171,172,179,180]
**MPV**	↑ in MDD [144,145,146,147], Depressive Symptoms [150], post-Ischemic Stroke depression [181,182], AD [175,183], PD [175], PanDis [177,184,185], GAD [146,173], ADHD [171,179,180,186]
	↓ in VaD patients [187] and in AD [187,188], MCI [188], PanDis [168,169,178]
	Negative correlation with PD progression stage (Hoehn and Yahr score [175,176]), Positive association with cognitive performance (MMSE) [187,188] although no correlation was reported elsewhere [175]
**PDW**	↑ in Depressive symptoms [150], PanDis [168,169,185], ADHD [171]
	↓ in VaD patients [187], AD [183,187,188], MCI [188]
	No association with PanDis [178], GAD [173]
**PCT**	↑ in MDD [145], ADHD [171,172,179,180,186]
	No association with GAD [173]
**Platelet serotonin metabolism**	↑ platelet serotonin receptor in MDD [189,190], Depressed CVD patients [191]
	↓ platelet serotonin transporter in MDD [192]
	↑ platelet serotonin uptake in AD [193]
	↓ platelet serotonin uptake in AD [194,195,196]
	No alteration in AD [197,198]
	↑ platelet serotonin levels in post-MI depressed patients [199] and Dementia [200]
	↓ platelet serotonin levels in AD [201,202,203] and ALS [204]
**Platelet mitochondria parameters**	↑ MAO-B activity in PD [205,206,207,208]
	↓ MAO-B activity in late phase AD [202]
	No alteration of MAO-B activity in PD [209,210]
	↓ mitochondrial Complex I activity in PD [211,212,213]
	No alteration of mitochondrial Complex I activity in PD [214,215]
	abnormal platelet mitochondrial morphology in ALS [216]
**Platelet activation markers**	↑ P-selectin in MDD [137,139], acute stress [143], AD [217]
	↑ platelet–leukocyte aggregates in acute stress [143], AD [217]
	↑ CD63+ platelets in MDD [139], chronic and acute stress [143]
	↑ PF4 and βTG in depressed CAD patients [218,219,220,221]

The studies reporting antidepressant drug-treated patients were excluded. ↑: increased or positive association; ↓: decreased or negative association. Plt: platelet count; MPV: mean platelet volume; PDW: platelet distribution width; PCT: plateletcrit. MAO-B: monoamine oxidase B; PF4: platelet factor 4; βTG: β-thromboglobulin. MDD: major depressive disorder, PanDis: panic disorder, GAD: generalized anxiety disorder, ASD: autism spectrum disorder ADHD: attention-deficit/hyperactivity disorder, MI: myocardial infarction, PD: Parkinson disease, AD: Alzheimer’s disease, MCI: mild cognitive impairment, CVD: cardiovascular disease, ALS: amyotrophic lateral sclerosis, CAD: coronary artery disease.

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
