# Peer review of "Beyond Haemostasis and Thrombosis: Platelets in Depression and Its Co-Morbidities"

_ijms, 2020, doi:10.3390/ijms21228817_

Round 1

Reviewer 1 Report

This is nice and comprehensive review on platelet function and MDD.

It would be nice if the authors could clearly state throughout the manuscript which of the effects of platelets are the result of depression-related treatments and which one to depression itself

Author Response

We thank the reviewer for her/his suggestion. Our major aim was to highlight the involvement of platelet pathophysiology in depression underlying possible intrinsic platelet characteristics that directly play a role in or are the consequence of depression. For this reason, the great majority of platelet functional studies reported in the review were performed on drug-free depressed patients or, when the study was longitudinal, only the findings relative to depressed patients before treatment were considered. In very few instances, we considered that also studies where the effect of anti-depressant was excluded by means of statistical analysis belonged to this first group of evidence (as an example see: Leake, A., et al., Studies on the serotonin uptake binding site in major depressive disorder and control post-mortem brain: neurochemical and clinical correlates. Psychiatry Res, 1991. 39(2): p. 155-65.) In fewer cases we also reported studies where the anti-depressant effect on platelets was investigated, mostly to elucidate platelet-dependent mechanisms important in depression or because these studies were the only ones available for certain platelet markers on relatively large cohorts (this was the case, for instance, for some studies that investigated platelet index distribution in association with depression or depressive symptoms). In any of the latter cases, the anti-depressant treatment was mentioned in the text. 

In order to follow the reviewer’s stuggestion and help the reader to understand our reasoning, we added a sentence at lines 50-52 as follows: ”With this review we first aim at providing functional evidence for a platelet pathophysiological involvement in depression by highlighting the intrinsic platelet characteristics in drug-free depressed patients.” In addition, we removed the first sentence of the “Epidemiological studies” subsection of the “MDD and platelets: further evidence of a link” section, reading: Anti-depressant drugs such as imipramine or trazodone have been first characterized as inhibitors of serotonin uptake by platelets [40].”

Reviewer 2 Report

Izzi et colleagues have developed an important comprehensive review for the potential role of platelets in patients with major depressive disorder. The review is nicely written, and I compliment the authors for their work. I would suggest the following comments to further improve this review so the readers would easier follow the available evidence so far for the link between platelet biomarkers and MDD.

Comments:

  1. Please revise the abstract to include more structured summary for the content of this review. The discussion of proteins linking platelets and neurons is missed in the abstract. Furthermore, as the authors conclude that there is lack of large studies addressing properly the relation between different platelet parameters and MDD, the suggestion that PDW is the best BM might be premature and too strong. The authors also comment that PDW is largely unknown from a functional perspective.
  2. I would suggest combining point 2 and 3 under the same subtitle, as the content of point 3 is a continuation of the point 2.
  3. The subtitles with points 7, 8 and 9 should come under point 6. “MDD comorbidities and platelets” to complete the paragraph.
  4. Under point 5, the authors discussed the role of gender on platelet-MDD link. I would suggest avoiding the term “gender” as it represents more than a biological difference between male and female individuals. Instead I would suggest to use “sex”. Interestingly the authors discuss the differential effect of sex in the association between platelet parameters and depression. Have the authors considered female-related factors in the regression models? It has been already demonstrated for female-related factors such as menstrual bleeding, hormonal/contraceptive therapy to importantly affect platelet parameters (ref. Blood. 2016 Jan 14;127(2):251-9).
  5. The authors should expand on the potential reasons/hypothesis for the observed differences between male and female individuals and the association between platelet parameters and MDD.
  6. Figure 1 and 2 are with poor resolution. Please revise the figures.

Author Response

Comment 1

Please revise the abstract to include more structured summary for the content of this review. The discussion of proteins linking platelets and neurons is missed in the abstract. Furthermore, as the authors conclude that there is lack of large studies addressing properly the relation between different platelet parameters and MDD, the suggestion that PDW is the best BM might be premature and too strong. The authors also comment that PDW is largely unknown from a functional perspective.

We thank the reviewer for this comment. We have changed the abstract following her/his suggestions.

Concerning the PDW, we agree with the reviewer that it might be too premature to consider it as the best biomarker of MDD. However, we still want to highlight its relevance as novel and present it as an interesting biomarker whose functional meaning in platelet biology is still largely unknown. Contrary to the other platelet indices (Plt and MPV), PDW variability does not seem to be strongly influenced by sex, and it is the only one presenting with significant genetic co-heritability with MDD. Based on these lines of evidence, we wish to still highlight its potential as a good candidate biomarker for MDD, also through the publication of Figure 2. The PDW represents the grade of heterogeneity in platelet size distribution of an individual and because platelet size has been described to have a role in platelet function and reactivity, this is an indication that it might have functional consequences. At the moment it is not known, however, what PDW variability exactly means in terms of platelet function, despite its increasingly recognized association with numerous clinical conditions apart from MDD. Another manuscript by our group, which is currently in preparation, specifically deals with the functional validation of PDW as a platelet activation marker, therefore we cannot speculate on that further in the present review. However, we can anticipate from our data that PDW variability is inversely associated with several platelet activation markers, therefore suggesting that its direct association with depression may be due to platelet activation status in MDD patients.

Based on all these elements, we have changed the abstract at lines 21-23 as follows: “Epidemiological and genetic studies discussed suggest that blood platelets might be relevant biomarkers of MDD prediction and occurrence also in the context of MDD comorbidities.” Later on in the manuscript we have made the following changes in the text:

  • at line 216-219: Although a direct link between PDW variability and platelet function has not yet been fully established, this evidence suggests once again that lower platelet activation and function may be a feature of depression, along with lower platelet and blood plasma serotonin, and lower platelet reactivity.”;
  • at lines 600-601: “Once again, PDW results as a novel and interesting candidate platelet biomarker for neuropsychiatric risk, although its functional and prognostic meaning remains to be elucidated.”
  • At lines 641-645: “Third, epidemiological and genetic studies increasingly identify PDW as one interesting yet functionally poorly understood candidate biomarker to test in terms of prediction of MDD and related neuropsychiatric/neurodegenerative risk in the future. The largely concordant evidence reported here that links PDW to MDD, ADHD, PanDis, as well as to cognitive traits and disorders like AD (Figure 2), is an incentive to further investigate this marker and its clinical implications.”

Comment 2

I would suggest combining point 2 and 3 under the same subtitle, as the content of point 3 is a continuation of the point 2.

Comment 3

The subtitles with points 7, 8 and 9 should come under point 6. “MDD comorbidities and platelets” to complete the paragraph.

We agree with the reviewer’s advice (comments 2 and 3) and have restructured the review accordingly.

Comment 4

Under point 5, the authors discussed the role of gender on platelet-MDD link. I would suggest avoiding the term “gender” as it represents more than a biological difference between male and female individuals. Instead I would suggest to use “sex”. Interestingly the authors discuss the differential effect of sex in the association between platelet parameters and depression. Have the authors considered female-related factors in the regression models? It has been already demonstrated for female-related factors such as menstrual bleeding, hormonal/contraceptive therapy to importantly affect platelet parameters (ref. Blood. 2016 Jan 14;127(2):251-9).

We thank the reviewer for her/his valuable comment. We have replaced the word “gender” with the word “sex” as suggested. Following the reviewer’s insightful suggestion, we built an enriched glm model within women, further adjusted for those variables available in Moli-sani which could influence their hormonal status, namely current menopause status, use of oral contraceptives and use of hormonal replacement therapy. Point 4 (lines 254-283) has been rewritten presenting this new analysis alongside with a new version of Table 1. The new applied models revealed results in line with our previous observations (Table 1), ruling out any potential confounding effect of these variables.

Comment 5

The authors should expand on the potential reasons/hypothesis for the observed differences between male and female individuals and the association between platelet parameters and MDD.

As explained above, we have performed a sex-specific analysis of platelet activation markers, including PDW in a dedicated cohort. These studies have highlighted different distribution of some activation markers, including PDW in men and women. At this moment, it is unclear whether sex is the only factor explaining differences for men and women in their association between platelet parameters and MDD, and a systematic analysis of further biological variables (e.g. sex hormone titers) – at the moment not available in Moli-sani - is needed.

We included an additional statement in our review at lines 279-283 as follows: "Of note, while potential confounders modifying hormonal status in women did not affect associations between platelet parameters and depressive symptoms within the Moli-sani study, sex hormones may still play a role in this association and further systematic analyses are warranted to rule out a potential influence of these variables, when hormone titers will be available within the cohort.”

Moreover, we added the following statement to the “Conclusions” section at lines 637-640: “Additional studies are warranted to verify whether the differential associations in women and men between platelet activation parameters and MDD results from intrinsic sex-specific (hormonal) differences influencing platelet activation or also from MDD-related factors.”

Comment 6

Figure 1 and 2 are with poor resolution. Please revise the figures.

Because inserting the figures in the word file (as recommended by the journal instructions) decreases their resolution, we are additionally sending the original figure files for the reviewer’s convenience.